# Green Synthesis of Silver Nanoparticles Using *Jacobaea maritima* and the Evaluation of Their Antibacterial and Anticancer Activities

**DOI:** 10.3390/ijms242216512

**Published:** 2023-11-20

**Authors:** Amal A. Althubiti, Samar A. Alsudir, Ahmed J. Alfahad, Abdullah A. Alshehri, Abrar A. Bakr, Ali A. Alamer, Rasheed H. Alrasheed, Essam A. Tawfik

**Affiliations:** 1Advanced Diagnostics and Therapeutics Institute, Health Sector, King Abdulaziz City for Science and Technology (KACST), Riyadh 11442, Saudi Arabia; aalthubiti@kacst.edu.sa (A.A.A.); abdualshehri@kacst.edu.sa (A.A.A.); aabakr@kacst.edu.sa (A.A.B.); aaalamer@kacst.edu.sa (A.A.A.); 2Bioengineering Institute, Health Sector, King Abdulaziz City for Science and Technology, Riyadh 11442, Saudi Arabia; salsadeer@kacst.edu.sa; 3Institute of Waste Management and Recycling Technologies, Sustainability & Environment Sector, King Abdulaziz City for Science and Technology, Riyadh 11442, Saudi Arabia; ajlfahad@kacst.edu.sa; 4Institute of Refinery and Petrochemicals, Energy and Industry Sector, King Abdulaziz City for Science and Technology (KACST), Riyadh 11442, Saudi Arabia; rrisheed@wafa.kacst.edu.sa

**Keywords:** *Jacobaea maritima*, silver nanoparticles, green synthesis, anticancer, antibacterial

## Abstract

Much attention has been gained on green silver nanoparticles (green-AgNPs) in the medical field due to their remarkable effects against multi-drug resistant (MDR) microorganisms and targeted cancer treatment. In the current study, we demonstrated a simple and environment-friendly (i.e., green) AgNP synthesis utilizing *Jacobaea maritima* aqueous leaf extract. This leaf is well-known for its medicinal properties and acts as a reducing and stabilizing agent. Nanoparticle preparation with the desired size and shape was controlled by distinct parameters; for instance, temperature, extract concentration of salt, and pH. The characterization of biosynthesized AgNPs was performed by the UV-spectroscopy technique, dynamic light scattering, scanning electron microscopy, X-ray diffraction, and Fourier-transform infrared. The successful formation of AgNPs was confirmed by a surface plasmon resonance at 422 nm using UV-visible spectroscopy and color change observation with a particle size of 37± 10 nm and a zeta potential of −10.9 ± 2.3 mV. SEM further confirmed the spherical size and shape of AgNPs with a size varying from 28 to 52 nm. Antibacterial activity of the AgNPs was confirmed against all Gram-negative and Gram-positive bacterial reference and MDR strains that were used in different inhibitory rates, and the highest effect was on the *E-coli* reference strain (MIC = 25 μg/mL). The anticancer study of AgNPs exhibited an IC_50_ of 1.37 μg/mL and 1.98 μg/mL against MCF-7 (breast cancer cells) and A549 (lung cancer cells), respectively. Therefore, this green synthesis of AgNPs could have a potential clinical application, and further in vivo study is required to assess their safety and efficacy.

## 1. Introduction

Infectious disease management is a major concern since some microbes show multi-drug resistance (MDR) against antibiotics, resulting in severe infection during complex operations, such as cardiac surgery and organ transplant, which can lead to increased mortality rates [1]. In addition, although the current chemotherapy strategy for cancer treatment is effective, the spread of the toxic effect on the surrounding healthy tissues and the chemo-resistance limits biomedical application [2]. The limitations in microbes and cancer treatment using antibiotics and chemotherapies emphasize the development of new treatment approaches, such as nanotechnology. Nanotechnology significantly accelerates the advancement of medical innovations with several special features for overcoming microbial antibiotic resistance [1]. Particularly, nanoparticles (NPs) possess remarkable properties, such as nano-size and charge, and exhibit significant anticancer and antimicrobial effects [3,4,5].

Among the different chemical, physical, and biological methods of nanoparticle synthesis, the biological synthesis of NPs using plants has been preferred over bacteria and fungi methods due to the fast process, toxicity elimination, energy consumption, and the obtained eco- and environment-friendly NPs [6]. Currently, various plant-mediated NPs were synthesized, including iron oxide NPs, copper NPs, calcium NPs, gold NPs, zinc oxide NPs, and silver NPs (AgNPs) using different plants such as *Hordeum vulgare* [7,8], *Phragmanthera austroarabica* extract [9], *Moringa oleifera* [10], *Spondias dulcis* [11], *Hibiscus sabdariffa* L. extract [12], and *Vaccinium archtostaphlyos* [13], respectively. In addition, AgNP applications in the food and healthcare industries have been investigated [14,15,16]. From an antimicrobial point of view, Ag has low toxicity and high antimicrobial activity in animal cells and was historically used in traditional medicine to overcome antimicrobial resistance and exert chronic wound and burn treatment [17,18]. Nevertheless, Ag in ionic forms, such as silver nitrate (AgNO_3_) and silver chloride (AgCl), have shown higher toxicity, including the repressed normal function of fibroblast in rats, induced hypersensitivity, and cardiac changes, such as left ventricular hypertrophy [19], while formed AgNPs were harmless and safe in biomedical application and hence stimulate the combination between traditional medicine and nanotechnology [19].

Enhancement of the antimicrobial effect of green NPs in comparison with physicochemical synthesized NPs has already been explored. This is due to the lack of toxic production when using biomolecules, such as plant metabolites, in the reduction and production of NPs [20]. Previous research has conducted a comparative study using chemical and biologically synthesized AgNPs on antibacterial activity and found that biosynthesized AgNPs resulted in higher antibacterial and antioxidant effects [21].

Recently, AgNPs have gained significant interest in the biomedical field due to their outstanding characteristics as a potential therapeutic approach in the treatment of different types of diseases such as bacterial and fungal infections, inflammation, and cancers. The green synthesis of AgNPs from natural sources has been demonstrated as significant in reducing the progression of human hepatic cancer cells. Applying the extract of a *Punica granatum* leaf, an ancient fruit belonging to the Punicaceae family, is reported to have antioxidant activity and free radical scavenging potency, suggesting promising and useful applications in the field of biomedical research [22] Moreover, the green synthesis of AgNPs from banana peel extract (BPE), which is an agricultural waste material that is used as a reducing and capping agent, revealed a cost-effective, non-toxic, and eco-friendly approach for the synthesis of AgNPs. Studies have shown a synergistic effect on the antimicrobial activity of the standard antibiotic levofloxacin against Gram-positive and Gram-negative bacteria [23]. Historically, plants have been utilized as a medicine by different cultures [18]. Their constituents, especially phenolic compounds, mediate the reduction and capping of metal ions and have therapeutic potential due to their antioxidant, anti-inflammatory, and antimicrobial properties [18,24]. Flavonoids are potent pro-oxidants stimulating the apoptotic pathways in cancer cells and behave as antioxidants in normal conditions [25]. Consequently, due to the properties of silver and plant metabolites for antimicrobial and anticancer, the biosynthesis of AgNPs using plant extract has emerged as a method of green synthesis [26]. *Jacobaea maritima,* in the subspecies sicula, belongs to the Asteraceae family and is sometimes called dusty miller [27,28]. It originated in Europe, the Mediterranean region [28]. This plant contains several minerals like phosphorus, iron, aluminum, calcium, magnesium, and potassium, and possesses different flavonoids, such as apigenin 7-O-glucoside, quercetin, dihydroquercetin, and luteolin 7-O-glucoside, and has the presence of compounds such as caryophyllene oxide, hydrocarbons, and distinct pyrrolizidine alkaloids [29,30,31]. Importantly, in traditional medicine, this plant is known for many medical benefits in various areas including antispasmodic [32], cataract [31], anxiety [32], migraine, and other severe conditions of eye-related such as opacity, conjunctivitis, and corneal clouding [31]. Antonella Maggio et al. reported the major constituents in the oil of *Jacobaea maritima* L., which include pentacosane, nonacosane, and heptacosane [29]. Therefore, *Jacobaea maritima* was selected in the current study for the biosynthesis of AgNPs due to its bioactive secondary metabolites’ components of flavones, amino acids, saponins, vitamins, tannins, alkaloids, aldehydes, and polysaccharides, which are well-known for their stabilizing, capping, and reducing properties for NP synthesis [33]. Hence, the aim of this study is the synthesis and characterization of AgNPs using an aqueous extract of *Jacobaea maritima* leaves. This study will evaulate anticancer activity against various cancer cell lines, as well as antibacterial activity against different bacterial strains.

## 2. Results and Discussion

### 2.1. Biosynthesis of AgNPs from Jacobaea maritima Extract

To obtain the optimal particle size of AgNPs, experiment parameters such as reaction time, precursor concentration, extract concentration, and temperature were adjusted and optimized. This optimized method resulted in effective AgNPs with a size of 37 ± 10 nm with a yellow color produced, as shown in Figure 1. Noticeably, the reaction time and temperature played important roles in obtaining the desired particle size. For instance, following a 24 h reaction time at ambient room temperature, AgNPs produced with 99 nm were compared to 156 nm in a 15 min reaction time at 100 °C and 37 nm in 72 h at a gradual temperature decrease from 22 °C to 4 °C (this has been used in subsequent experiments). The desired AgNP size is identical to previous research that has used Skimmia laureola for AgNP synthesis, which has a spherical shape and an average size of 38 ± 0.27 nm and antimicrobial activities against *S. aureus*, *P. vulgaris*, *P. aeruginosa,* and *E. coli* [34,35]. AgNP biosynthesis with a size range from 22 nm to 52 nm was also confirmed using *Croton sparsiflorus Morong* leaf extract [36]. Another study using leaf extract of Justicia spicigera for the green synthesis of AgNPs has shown a 100 nm size when using 60 °C as a reaction temperature [37]. Table 1 shows the previous literature synthesizing AgNPs from distinct plant extracts.

The mechanism of AgNP formation using plant extract is illustrated by the crucial roles of proteins and secondary metabolites, such as polyphenols and flavonoids (quercetin), as reduction and stabilizing agents for AgNP formation [41]. Active compounds present in *Jacobaea maritima* had already been identified by Voynikov et al. using Ultra-High-Performance Liquid Chromatography–High-Resolution Mass Spectrometry (UHPLC-HRMS) and are summarized in (Appendix A). These compounds are classified as either flavonoids or hydroxybenzoic, hydroxycinnamic, and acylquinic acids along with their derivatives, with flavonols such as quercetin, kaempferol, and isoquercitrin being the highest presented components [30]. Certain functional groups are identified for a reduction in Ag ions and metallic Ag, including hydroxyl groups attached to aromatic rings and carboxyl groups, which are present in the active compound. It is plausible that the enol–keto tautomeric transformation of these compounds could release a reactive free electron that reduces silver ions to metallic silver [41].

Once these metabolites interact with the silver ions, nanoparticle production occurs, and this process of formation is composed of three main steps: ions reduction to metal atoms, atom clustering and those results in small clusters, and their growing and formed NPs protected by capping agents to prevent aggregation. These steps all depend on different factors, such as the concentration of stabilizing and reducing agents of plant extract, reaction temperature, speed of stirring, pH of the medium, and precursor concentration (i.e., AgNO_3_) [41]. In addition, these factors also play essential roles in the charge, sizing, and polydispersity of the prepared nanomaterials [42]. The gradual decrease in the temperature of the process of AgNP formation from 22 °C to 4 °C in a reaction duration within 72 h resulted in a small AgNP size, and it has already been established that the reaction temperature affects the formation and growth of AgNPs; the higher the temperature, the larger the particle size [43], which agrees with the obtained results. Our plausible explanation is that high temperatures accelerate the nucleation and growth of NPs, which occurs via Ostwald ripening or coalescence to form larger particles that are more energetically stable [44].

This time difference in the AgNP formation might likely be due to the extraction condition and method, solvent type, and concentration of leaf extract metabolites. Importantly, five main factors have influenced the biosynthesis of AgNPs: reductant concentration (extract), temperature, reaction time, and AgNO_3_ concentration and its ratio to the plant extract, following a previous study by Pungle et al. [1].

### 2.2. Ultra-Visible Spectroscopy of AgNPs

Biosynthesized AgNPs from *Jacobaea maritima* extract were generated through a reduction in Ag^+^ ions, which was observed by the UV-visible spectra at 422 nm, indicating spherical nanoparticle formation [45]. The UV peak of the plant extract showed a wavelength of 323 nm (Figure 2), which may be related to the organic compounds in the plant leaves, and this is in agreement with a previous observation of UV results by Pungle et al. and Devavesan and Alsalhi [1,45], whose synthesized AgNPs were in a wavelength range of 400 and 435 nm, reflecting the AgNP shape and size [1].

### 2.3. Dynamic Light Scattering (DLS)

AgNPs demonstrated average diameters of 37 nm, a polydispersity index (PDI) of 0.225, and a zeta potential value of −10.9 ± 2.3 mV (Table 2). It was found that by varying the reaction temperatures during AgNP synthesis, from 100 °C to the ambient temperature at 22 °C until 4 °C, the NP particle size gradually decreased from 156 to 99 to 37 nm, respectively (Table 2). Salayová et al. reported similar observations of the diameter size and zeta potential results for the AgNPs synthesized from *Origanum vulgare* [39]. Flieger et al. reported green synthesis of AgNPs from natural extracts at a 90 °C reaction temperature, and with increased incubation time in this temperature, the hydrodynamic diameter was elevated from 99 nm to 119 nm and the zeta potential decreased from −9 to −56 mV [46]. The smaller the NPs, the larger their surface area; and, therefore, the density of capping agents increases on the surface. Owing to the enol–keto tautomeric transformation of compounds present in the plant extract, which releases a reactive free electron to reduce silver ions to metallic silver, most capping agents present in the solution will lose their charges, thereby reducing the zeta potential value of formed AgNPs. This is confirmed by FTIR analysis that shows the loss of OH stretching vibration at 3239 cm^−1^ for AgNPs while having stronger peaks at 1032 cm^−1^ and 1261 cm^−1^, which can be attributed to the C-O stretching of carboxylic acid, esters, or ethers.

### 2.4. Scanning Electron Microscope (SEM)

The morphology of the AgNPs was exhibited as a spherical shape with a size varying from 28 to 52 nm (Figure 3), which is similar to previously reported SEM for AgNPs synthesized from *Hypericum perforatum* L. extract [41].

### 2.5. Fourier-Transform Infrared Spectroscopy (FTIR)

FTIR was utilized to identify the biomolecule components of the extract that might behave as reducing and capping agents to reduce silver ions to AgNPs. It is advantageous to use plant extract in AgNP synthesis, as multiple capping agents can form a thin layer surrounding the NPs [26]. The FTIR spectra exhibited bands at 3239, 2929, 1597, 1265, 1041, and 861 cm^−1^ for the *Jacobaea maritima* extract and 2965, 2079, 1653, 1261, 1032, and 803 cm^−1^ for biosynthesized AgNPs, as shown in Figure 4. These results indicate the functional groups responsible for stabilizing and a reduction in AgNPs using an extract of *Jacobaea maritima*. In the AgNP spectrum, the peak at 803 cm represents the =CH group in aromatic compounds [47], and the peak at 1032 cm^−1^ was attributed to the C-O functional group of carboxylic acid [46], while the band at 1261 cm^−1^ shows phenolic group presence and is attributed to the stretching of O-H. The observed peak at 1653 cm^−1^ corresponds to alkenes and aromatic rings and is attributed to C=C bond stretching [48]. The presence of NH^2+^/NH^3+^ in the peptide bonds was shown by the sharp peak at 2079 cm^−1^, while the medium single-bonded C-H alkane group was assigned by the spectral peak at 2965 cm^−1^. Results obtained mimic previous research showing the plant extract active molecules that behave as capping and reducing agents of Ag^+^ to Ag^0^ [45,49]. Based on these results, compounds containing carbonyl and phenolic groups along with aromatic rings can be found on the surface of NPs, which is evidence of the presence of flavonoids or acid extract along with their derivatives. Thereby, FTIR revealed the efficient capping and stabilization of NPs.

### 2.6. X-ray Diffractometer (XRD)

The XRD pattern showed distinct diffraction peaks at 31.84°, 37.81°, 43.9°, 64.2°, and 77.25°, which corresponded to 111, 200, 220, and 311 planes, respectively (Figure 5). The distinct diffraction peak of the XRD at 38.24°, which is related to the 111 planes, indicated the purity of the prepared AgNPs. This was in agreement with the XRD results of a published study by Vanaja and Gurusamy, who synthesized AgNPs from *Coleus aromaticus* leaf extract [50]. The previous report has illustrated that this Bragg reflection plane relates to the silver face-centered cubic (fcc) phase. The crystallite size of AgNPs was calculated by the Scherrer formula, as previously reported [51], with similar XRD results as Vanaja and Gurusamy [50]:

$D=\frac{Kƛ}{\beta cos\theta }$ = 47.4 nm, which is between the range of 28 nm and 52 nm that was reported in the SEM results.

### 2.7. Anticancer Activity of Biosynthesized AgNPs

The cytotoxic effect of biosynthesized AgNPs on MCF-7 (Figure 6A) and A-549 (Figure 6B) cells was demonstrated in a dose–dependent manner in the range of 0.35 μg/mL, 0.7, 1.4, 2.7, and 5.5 after 24 h of exposure. The leaf extract did not show a cytotoxic effect in the same concentrations of AgNPs that were exposed to both cells. The results showed a decrease in the viability of the MCF-7 and A-549 cells, and its inhibitory response was observed at a concentration of ≥1.4 µg/mL. More than 80% of cell inhibition occurred at the highest concentration, i.e., 5.5 μg/mL, compared with 1.4 μg/mL. The IC_50_ values of biosynthesized AgNPs against MCF-7 (1.37 μg/mL) and A-549 (1.98 μg/mL) confirmed that NPs exhibited a tumor-inhibitory effect higher than the previously synthesized particles from *Hypericum perforatum* L. [41].

AgNPs generally inhibit cancer cells primarily by the plant extract surface coating as an external stimulus, inducing oxidative stress by the free radical production and thus suppressing tumor gene’s function, mitochondria-dependent pathways, and lipid peroxidase induction [52]. Another proposed cause of cell apoptosis using AgNPs is by triggering an increase in intracellular Reactive Oxygen Stress (ROS), and reactive hydroxyl radical damage DNA [52]. These could lead to degenerative changes and mitochondrial dysfunction in the expression of some pro-apoptotic (*Bax*) and anti-apoptotic (*Bcl*-2) genes and mitochondria proteins, such as releasing Cyt C, a pro-apoptotic mitochondrial protein, into the cytosol and apoptotic cellular induction by caspase-3 and caspase-9 activation proteins [42,53,54]. Induction of the ROS occurs through the production of the reactive oxygen species in a concentration that can inhibit antioxidant enzymes, including glutathione and superoxide dismutase eliminate antioxidants, and, consequently, damage the DNA [55]. Furthermore, according to recent research, AgNPs synthesized from a plant are shown to trigger p53 protein activation, which suppresses the formation of malignant tumors in mammalian cells [56]. It could be inferred from these results that biosynthesized AgNPs have a valuable reduction impact on cancer cells with a low inhibitory concentration, and further in vivo examination might be performed to prove their future anticancer treatment potential in humans.

### 2.8. Antibacterial Activity of Biosynthesized AgNPs

Silver and its compounds are mostly known to have antibacterial effects against a wide range of microorganisms [57]. The antibacterial results (i.e., MIC) showed that all tested bacterial strains were not inhibited by the *Jacobaea maritima* extract at the same concentrations of tested AgNPs, as shown in Table 3 and Appendix A, and these bacterial activities concurred with previous studies by Alahmad et al. and Asif et al. using the extract of *Carduus crispus* and *Hypericum perforatum* [58,59]. Furthermore, all the isolates were inhibited using AgNO_3_ at 25 µg/mL, except for the antibiotic-sensitive *E. coli* strain (ATCC 25922) that displayed an MIC value of 6.25 µg/mL, as shown in Table 3 and Appendix A. The AgNPs demonstrate results combined with the distinct sensitivity-resistance levels of the synthesized AgNPs against different strains, illustrating that bacteria strains are sensitive mainly due to AgNPs and not because of silver ions.

Biosynthesized AgNPs possessed antibacterial activity against Gram-positive and Gram-negative referenced bacteria and clinical isolates a various inhibitory rate. In general, AgNPs inhibited both sensitive and methicillin-resistant *S. aureus* (ATCC 29213 and ATCC 43300) at an MIC of 100 µg/mL, while *S. epidermidis* (ATCC 12228) has an MIC of 50 µg/mL, which is half the clinical isolate (isolate 5029; MIC of 100 µg/mL). The AgNPs exhibited an MIC of 25 µg/mL against *E. coli* (ATCC 25922), which is a quarter of the clinical isolate (isolate 1060, MIC of 100 µg/mL). The MIC of *P. aeruginosa* (ATCC 27853) was 50 µg/mL, which is half the clinical isolate (isolate 7067; MIC of 100 µg/mL). Owing to the smaller size of AgNPs (i.e., 37 ± 10 nm), higher surface area and more silver ions will be released, resulting in antibacterial activity [60]. These results are consistent with previous research utilizing AgNPs of *Pedalium murex* and *Sisymbrium irio* leaf extracts [61,62], and the low activity against Gram-positive is found in a previous study [39].

It is apparent that the antibacterial activity of the AgNPs on Gram-negative reference strains was higher than Gram-positive strains, and this can be attributed to the AgNP attachment and accumulation on the bacteria cell surface, causing alteration and denaturation of the cell membrane structure. *S. aureus* is the most resistant Gram-positive bacterial strain, and this could be due to the morphological structure of the cell wall [63]. This observation was consistent with a previous study that demonstrated a higher sensitivity of *E. coli* to AgNPs compared to *S. aureus* owing to the thicker layer of peptidoglycan that is a unique structural feature of Gram-positive bacterial cell walls [60]. On the other hand, the higher susceptibility of a Gram-negative bacterium to AgNPs was due to easier penetration through its cell wall, which is composed of a thinner layer of peptidoglycan compared to *S. aureus* [64]. The concentration of the AgNPs may also play a vital role in the antibacterial effect, as reported by Sankar et al. [65]. This study confirmed that by increasing the concentration of AgNPs against different bacterial strains from 0.78 to 100 μg/mL, bacterial inhibition occurred, and no bacterial inhibition was observed at <25 µg/mL.

## 3. Materials and Methods

### 3.1. Materials

The cell lines used in the current work were purchased from the American Type Culture Collection (ATCC, Manassas, VA, USA). AgNO_3_ and Dulbecco’s modified eagle medium (DMEM) and its supplements were obtained from Sigma-Aldrich (St. Louis, MO, USA). 3-(4,5-dimethylthiazol-2-yl)-5-(3-carboxymethoxyphenyl)-2-(4-sulfophenyl)-2H-tetrazolium (MTS), which is also known as a cell Titer 96^®^ aqueous one solution cell proliferation assay, was supplied by Promega (Southampton, UK). *Staphylococcus aureus* (*S. aureus*, ATCC 29213), *Staphylococcus epidermidis* (*S. epidermidis*, ATCC 12228), *Escherichia coli* (*E. coli*, ATCC 25922), *Pseudomonas aeruginosa* (*P. aeruginosa*, ATCC 27853), and methicillin-resistant *S. aureus* (MRSA, ATCC 43300) were all purchased from ATCC as reference bacteria. Other bacterial strains were clinically isolated (multi-drug-resistant) MDR bacteria, which include *E. coli* (isolate 1060), *P. aeruginosa* (isolate 7067), and *S. epidermidis* (isolate 5029). Mueller–Hinton broth (MHB) was obtained from Scharlau (Barcelona, Spain) and prepared according to the manufacturer’s instructions. A Milli-Q^®^ IQ 7005 Purification System (Millipore SAS, Molsheim, France) was used to generate distilled water.

### 3.2. Plant Collection, Classification, and Green Synthesis of AgNPs

*Jacobaea maritima* leaves were commercially purchased from a local plantation in Riyadh, Saudi Arabia (Figure 7). The plants were authenticated by Dr. Ali A. Namazi, who is the General Manager of Advanced Agricultural and Food Technologies Institute at King Abdulaziz City for Science and Technology (KACST), Riyadh, Saudi Arabia. Leaf extraction was prepared firstly by rinsing with tap water followed by distilled water three times to remove all dust and unwanted particles. About 25 gm of finely incised *Jacobaea maritima* leaves were boiled in 300 mL of distilled water at 100 °C for about 20 min, followed by decreasing the temperature to 50 °C for 25 min. Next, the extract was left to reach room temperature and was followed by filtration to eliminate particulate matter using Whatman filter paper No. 1. A pale light-brown solution was obtained and stored at 4 °C for AgNP synthesis.

For AgNP synthesis, AgNO_3_ was used as a precursor following the method of AgNP synthesis using *Hypericum perforatum* L. [41] with slight modification. Briefly, an AgNO_3_ solution was prepared at a concentration of 1mM, while the extract was prepared by mixing 25 gm of finely incised leaves with 300 mL of distilled water, and the mixture was left to boil for 20 min. Then, the temperature was decreased to 50 °C and kept for 25 min. Next, filtration was carried out to obtain the extract solution. Finally, a 1:1 ratio of an AgNO_3_ extract solution was used and placed on a magnetic stirrer at 220 rounds per minute at ambient temperature for a specific time, followed by a gradual temperature decrease for 72 h. Color change from light yellow to light brown is indicative of AgNP formation.

### 3.3. Ultraviolet–Visible (UV/Vis) Analysis for AgNP Characterization

UV/Vis spectroscopy absorption of AgNPs synthesized in an aqueous solution was measured using a LAMBDA 850+ UV/Vis spectrophotometer (PerkinElmer, Waltham, MA, USA) at wavelength range from 800 to 300 nm to obtain the lambda maximum (ƛ max) of AgNPs. Distilled water was used as an experimental control.

### 3.4. Dynamic Light Scattering (DLS)

The AgNP hydrodynamic diameter was determined via DLS, which is used to evaluate the particle size, polydispersity index, and the particle’s electrical charge, i.e., zeta potential. The measurements were conducted using Zetasizer (NANO ZSP, Malvern Panalytical Ltd., Malvern, UK).

### 3.5. Scanning Electron Microscopy (SEM)

SEM (JSM-IT500HR SEM, JEOL Inc., Peabody, MA, USA) was used to evaluate the morphological characterization of the AgNPs. In preparation for imaging, we deposited a drop of the nanoparticle suspension on an SEM stub and allowed it to dry at ambient temperature. Then, the results were analyzed utilizing the microscope’s software (SEM Operation: 3.010, version 1.010, JOEL TECHNICS Ltd., Tokyo, Japan).

### 3.6. Fourier-Transformed Infrared Spectroscopy (FTIR)

FTIR measurements were conducted to analyze the functional groups on the AgNP surface via a wavelength range of 4000–400 cm^−1^ using an Agilent Cary 630 ATR-FTIR analyzer (Agilent Technologies Inc., Santa Clara, CA, USA).

### 3.7. X-ray Diffraction (XRD)

XRD analysis was conducted using an ARL Equinox 1000 X-ray diffractometer from Thermo Scientific (Japanska, New Belgrade, Serbia) to determine biosynthesized AgNP crystalline structure. The analysis was operated using a Cu Kα radiation source (ƛ= 1.54 Å) at a voltage of 40 kV, a current of 30.5 mA, and a full 2θ range of 3–100° for 60 min. Materials were prepared using the powder of an AgNP colloidal solution that was freeze-dried, proving the fcc lattice structure for AgNPs that is synthesized from *Jacobaea maritima* (JCPDS: File No. 96-901-3046).

### 3.8. In Vitro Cytotoxicity Assessment

Cell viability was tested utilizing a standard MTS assay after exposure to biosynthesized AgNPs and *Jacobaea maritima* leaf extract for 24 h following a modified method of previous studies [66,67]. Two cancer cell lines were used: a breast cancer cell line (MCF-7, ATCC HTB-22), and a lung cancer cell line (A549, ATCC CCL-185). Cells were routinely maintained in DMEM supplemented with 10% (*v*/*v*) fetal bovine serum (FBS), streptomycin 10 mg/mL, 1% L-glutamine, and penicillin 100 U/mL. Initially, cells were harvested using trypsin and counted with the trypan blue exclusion test, followed by seeding in a 96-well plate at a density of 1.5 × 10^4^ cells/well, followed by overnight incubation at 37 °C and 5% CO_2_. Then, cells were treated with AgNPs (stock concentration= 13.18 µg/mL) utilizing a concentration range of 0.35 µg/mL, 0.7 µg/mL, 1.4 µg/mL, 2.7 µg/mL, and 5.5 µg/mL for 24 h. Cells incubated with DMEM and 0.1% triton x-100 only were used as the positive and negative control, respectively. Subsequently, 100 µL of DMEM was added after removing the samples from the wells. Then, 20 µL of the MTS reagent was applied into each well and incubated for a further 3 h at 37 °C. After that, the Cytation 3 absorbance microplate reader (BIOTEK Instruments Inc, Winooski, VT, USA) was used to measure the absorbance at 492 nm. Lastly, the cell viability% was calculated using the following formula:Cell Viability % = (S − T)/(H − T) × 100(1)
where S is the absorbance of the AgNP-treated cells, T is the absorbance of the triton x-100-treated cells (negative control), and H is the absorbance of the cells treated with DMEM (positive control).

### 3.9. The Minimum Inhibitory Concentration (MIC)

The MIC was utilized to evaluate the antibacterial activity of *Jacobaea maritima* extract AgNPs and AgNO_3_ following the method used by Singh and Mijakovic [68] with slight modifications and the CLSI reference method [69]. A serial dilution of (100 to 0.78 μg/mL) was added into the 96-well microtiter plates containing MHB at an ultimate volume of 100 μL in each well. Then, single pure colonies of the bacterial inoculums (100 μL), adjusted to the 0.5 McFarland standard, were added to each well to give a cell density of 1.5 × 10^6^ CFU/mL. The 96-well microtiter plates were incubated overnight at 37 °C with a continuous shaking speed of 140 rounds per minute (RPM). The lowest concentration with no growth (no turbidity) is considered the MIC. Additionally, bacterial growth inhibition was measured using a PowerWave XS2 plate reader (bioMérieux, Marcy L’Etoile, France) at a UV absorbance of 600 nm. The wells containing only bacteria and media were used as positive and negative controls, respectively.

### 3.10. Statistical Analysis

Experiments were performed as three independent replicates, and the results were expressed as the mean ± standard deviation (SD) and calculated using Microsoft Excel 2022. The results of UV/Vis, FTIR, and XRD were plotted using OriginPro 2021 software (OriginLab Corporation, Northampton, MA, USA).

## 4. Conclusions

This study demonstrated a successful biosynthesis of AgNPs using *Jacobaea maritima* leaf extract as a capping and reducing agent, which possesses a narrow size and spherical shape and potent antibacterial and anticancer efficacies. This successful AgNP synthesis was achieved by altering one factor at a time in the optimization process including plant and AgNO_3_ concentrations, reaction temperature, and duration. Different characterization techniques were used. The DLS demonstrated the average particle size and zeta potential of the prepared AgNPs of 37 ± 10 nm and −10.9 ± 2.3 mV, respectively, while the SEM further confirmed their dominant spherical shape with size range from 28 to 52 nm. The UV/vis spectroscopy confirmed the formation of nano-sized AgNPs with wavelength spectra at 422 nm and color change observation (yellow-colored nanoparticle suspension). The XRD confirmed the crystallite nature of AgNPs with a theatrical AgNP size of 47 nm. FTIR showed the functional group of the leaf extract responsible for the reduction and capping of the synthesized AgNPs. Biosynthesized AgNPs exhibited antibacterial activity in a concentration-dependent manner and were highly effective against Gram-negative bacterial strains, with the MIC value for *E. coli* bacterium being the lowest (25 μg/mL). Promising anticancer property of biosynthesized AgNPs was shown against breast cancer (i.e., MCF-7) and lung cancer (i.e., A-549) cell lines, with an IC_50_ of 1.37 μg/mL and 1.98 μg/mL, respectively. Future in vivo studies are required to evaluate the effectiveness and safety of biosynthesized AgNPs as an alternative conventional anticancer and antibacterial agent.

## Figures and Tables

**Figure 1 ijms-24-16512-f001:**
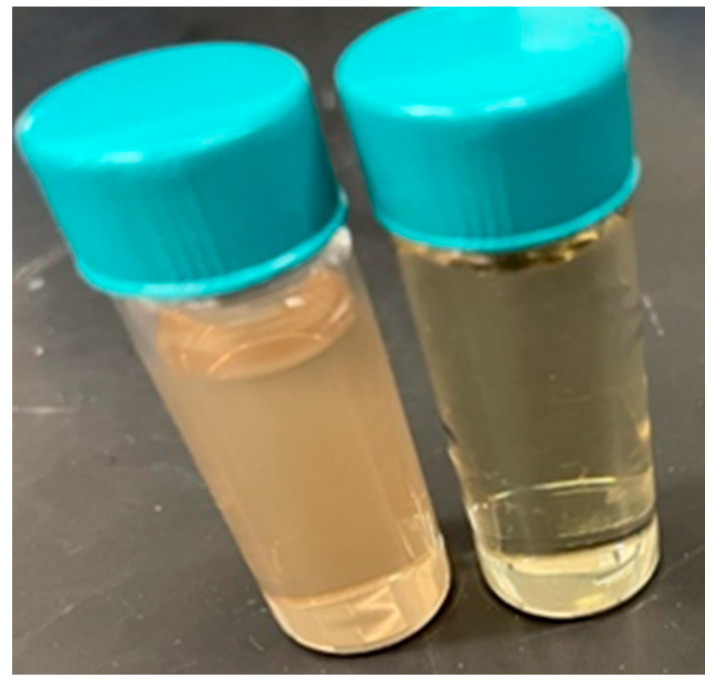
The color of *Jacobaea maritima* extract (**right**) and biosynthesized AgNPs (**left**).

**Figure 2 ijms-24-16512-f002:**
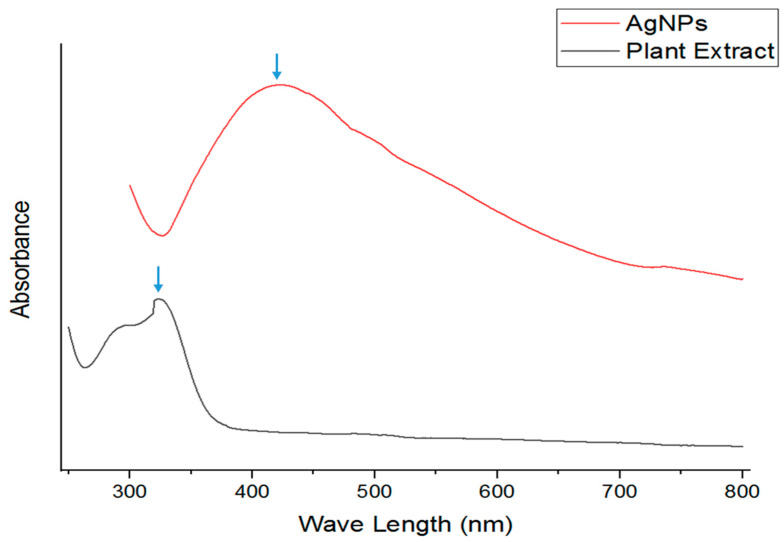
The UV absorbance peak of *Jacobaea maritima* leaf extract and biosynthesized AgNPs were shown at wavelengths of 323 nm and 422 nm (blue arrows), respectively.

**Figure 3 ijms-24-16512-f003:**
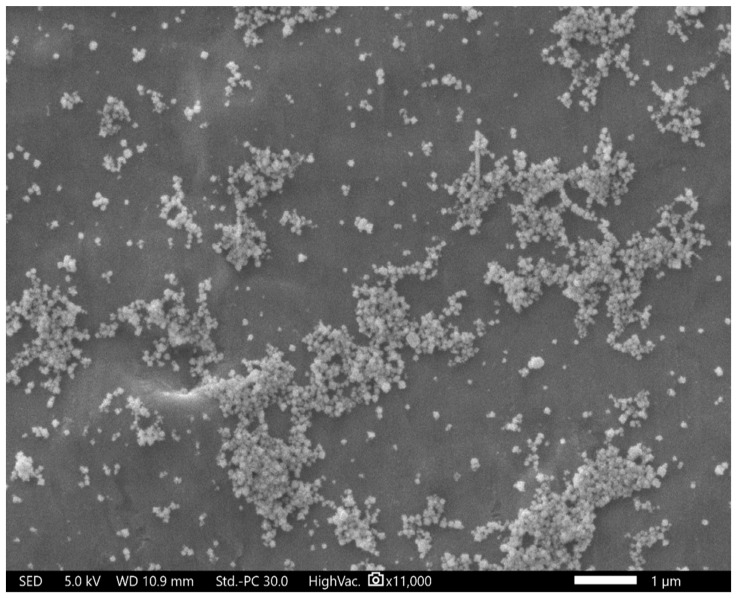
SEM image at 37,000 magnifications showing the spherical shape of AgNPs and a size range between 28 nm and 52 nm.

**Figure 4 ijms-24-16512-f004:**
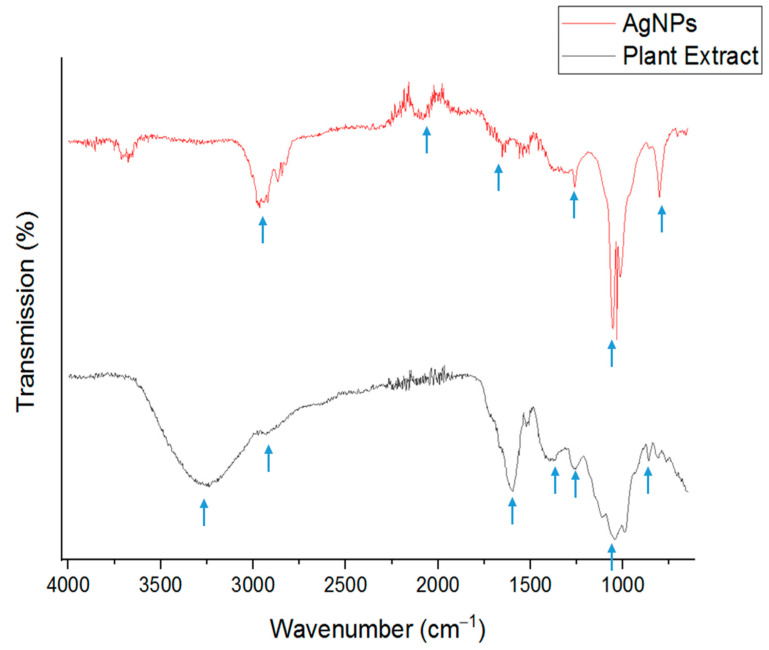
FTIR spectra of *Jacobaea maritima* and biosynthesized AgNPs in the wavelength range of 4000–400 cm^−1^ showing the functional groups within the extract responsible for a reduction in Ag^+^ to NPs. The blue arrows represent the distinctive peaks.

**Figure 5 ijms-24-16512-f005:**
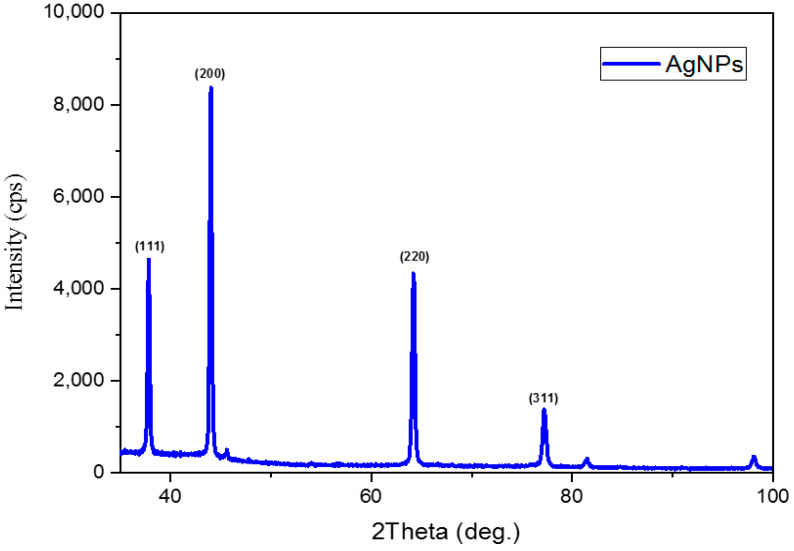
XRD of biosynthesized AgNPs confirms the crystalline nature of NPs.

**Figure 6 ijms-24-16512-f006:**
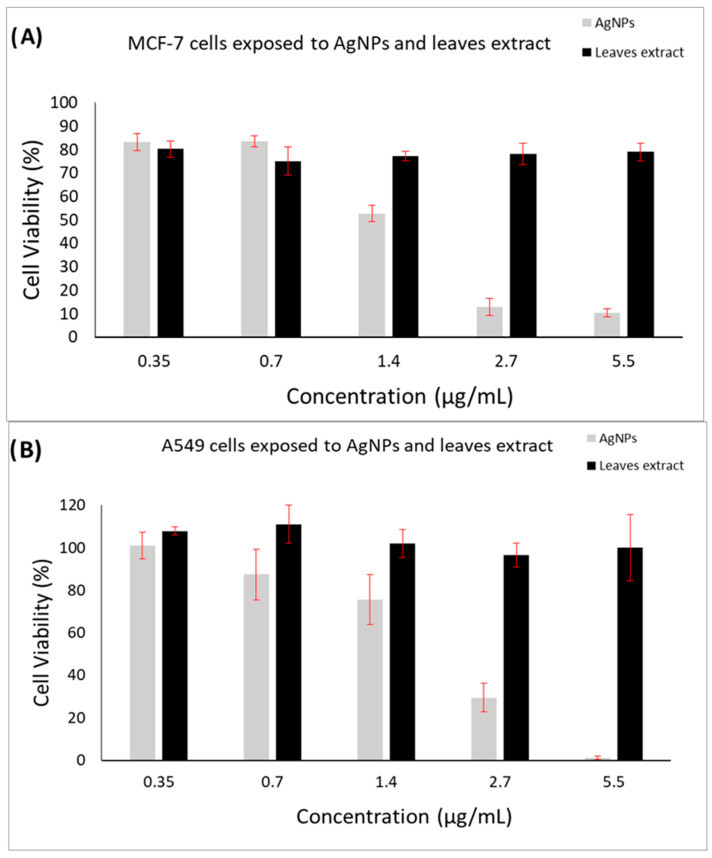
Cell viability of biosynthesized AgNPs and *Jacobaea maritima* leaf extract on (**A**) MCF-7 and (**B**) A-549 after 24 h of exposure to the cells. These data are the MTS assay results, which are expressed as cellular viability (%) and presented as the mean ± SD (*n* = 3).

**Figure 7 ijms-24-16512-f007:**
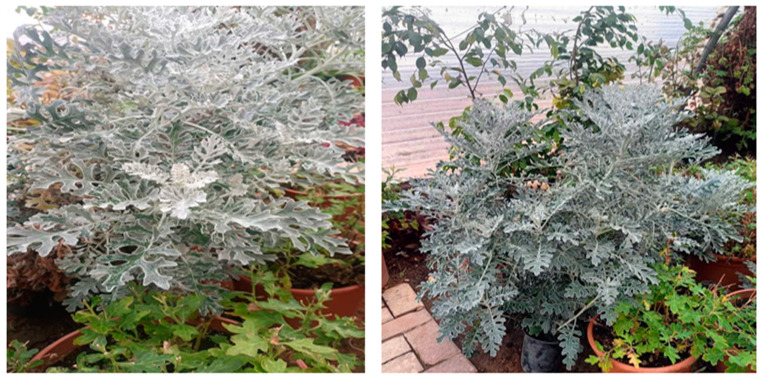
*Jacobaea maritima* leaves.

**Table 1 ijms-24-16512-t001:** A summary of relevant findings of biosynthesized AgNPs using various extracts.

Extract	Reaction Temperature	Reaction Time	AgNP Size	Reference
*Skimmia laureola*	23 °C	10 min	38 ± 0.27 nm	[35]
*Croton sparsiflorus Morong*	29 °C	NA	22 nm–52 nm	[36]
*Justicia spicigera*	60 °C	15 min	86 nm–100 nm	[37]
*Solanum tricobatum*	37 °C	24–48 h	52 nm	[38]
*Datura stramonium*	80 °C	2 h	13 nm–60 nm	[21]
* Origanum vulgare * *Lavandula angustifolia* * Capsella bursa-pastoris *	80 °C	48 h	46.1 ± 19.7 nm 37.8 ± 10.7 nm 16.2 ± 8.4 nm	[39]
* Achillea maritima * subsp. *maritima species*	NA	NA	14 nm–21 nm	[40]
*Jacobaea maritima*	4 °C	72 h	28 nm–52 nm	(This study)

**Table 2 ijms-24-16512-t002:** Dynamic light scattering (DLS) and the zeta potential of biosynthesized AgNPs in different reaction temperatures. The blue data are the ones used as the successfully synthesized AgNPs.

Temperature	Size (nm) ± SD	PDI ± SD	Zeta Potential ± SD
100 °C	156 ± 4	0.295 ± 0.009	−31.7 ± 4.6
Ambient RT (22 °C)	99 ± 4	0.273 ± 0.037	−33.8 ± 8.9
22 °C–4 °C	37 ± 10	0.225 ± 0.006	−10.9 ± 2.3

**Table 3 ijms-24-16512-t003:** MIC values of the *Jacobaea maritima* extract and AgNPs against different bacterial strains. The results are presented as mean ± SD of triplicates. (-) indicates no inhibition of bacterial strains.

Microorganisms	MIC (μg/mL)	
	*Jacobaea maritima*	AgNPs
*S. aureus* (ATCC 29213)	-	100 ± 0
*MRSA* (ATCC 43300)	-	100 ± 0
(ATCC 12228) *S. epidermidis*	-	50 ± 0.01
(isolate 5029) *S. epidermidis*	-	100 ± 0.01
(ATCC 25922) *E. coli*	-	25 ± 0
(isolate 1060) *E. coli*	-	100 ± 0.01
(ATCC 27853) *P. aeruginosa*	-	50 ± 0.01
(isolate 7067) *P. aeruginosa*	-	100 ± 0

## Data Availability

The authors confirm that the data supporting the findings of this study are available within the article.

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
