# Peer review of "Green Synthesis of Silver Nanoparticles Using Jacobaea maritima and the Evaluation of Their Antibacterial and Anticancer Activities"

_ijms, 2023, doi:10.3390/ijms242216512_

Round 1
Reviewer 1 Report
Comments and Suggestions for Authors
"The manuscript is interesting, original and well presented. I recommend the mention MINNOR ERRORS. I list below my comments:
1. Page 3, line 97-100: Can the authors discuss their findings regarding the size of AgNPs in comparison to the existing literature?"
2. Page 5, figure 3: Could the authors consider replacing the SEM image with one that better visualizes the spherical shape of AgNPs for improved clarity?
3. Page 6: line 178: It is important to mention the miller index of the crystallite size that was calculated?
4. Page 6 , line 170-171, page 7 and figure 5: Could the authors identify the two peaks located at approximately 80° and at 100°?
5. Page 11, line 317: please replace particles’ by particle’s."
Author Response
We thank the reviewers for their thoughtful and helpful comments. We respond to each point below; their suggestions have undoubtedly improved the paper, for which we are very grateful. All changes in the revised manuscript were in yellow highlighted.
- Page 3, line 97-100: Can the authors discuss their findings regarding the size of AgNPs in comparison to the existing literature?"
Response: Thank you for your suggestion. Our findings are comparable with previous studies (Summarized in Table 1, Page 4 in the revised manuscript), in terms of particle size, yet we utilized a lower reaction temperature at the expense of a longer reaction time.
- Page 5, figure 3: Could the authors consider replacing the SEM image with one that better visualizes the spherical shape of AgNPs for improved clarity
Response: A higher resolution SEM image has been provided with a better visualization of the spherical shape of AgNPs (Figure 3, Page 6 in the revised manuscript).
- Page 6: line 178: It is important to mention the miller index of the crystallite size that was calculated?
Response: The miller index of the crystalline size has been provided in the revised manuscript (Figure 5, Page 8 in the revised manuscript, XRD of the biosynthesized AgNPs confirms the crystalline nature of the NPs).
- Page 6, line 170-171, page 7 and figure 5: Could the authors identify the two peaks located at approximately 80° and at 100°?
Response: Thank you for pointing this out. The peak is located at approximately 80°C = 81.4, and the peak near 100°C = 98.01.
- Page 11, line 317: please replace particles’ by particle’s.
Response: This has been replaced (Page 12, Line 366 in the revised manuscript).
Reviewer 2 Report
Comments and Suggestions for Authors
The manuscript “Green Synthesis of Silver Nanoparticles Using Jacobaea Maritima and the Evaluation of its Antibacterial and Anticancer Activities”, studied a method for synthesis of AgNPs using extract of Jacobaea Maritima leaves, then evaluated the anticancer and antimicrobial activity of resultant AgNPs. Overall, none of the methods and properties have sufficient novelty based on the literature review in the introduction.
I will consider recommending the manuscript to IJMS if the authors can provide superiority in either approach or properties over other reports.
There are a few questions that might be helpful to better understand the mechanism for readers if the authors like to address them kindly within this manuscript.
- What are the components in the extraction of Jacobaea Maritima leaves, and their concentration?
- Which component is involved in the reduction reaction with AgNo3, and what is pathway of reaction?
- What are the concentrations of active extraction and silver nitrate in line 303-305?
- Why can the lower reaction temperature result in smaller AgNPs?
- what component played as capping agents of resultant clusters, and why did the zeta potential turn from negative (-31.7) to positive (10.9)?
- The SEM image is not clear at all, using a TEM image is better to measure the particle size as in your previous paper.
- I can’t find the column indicating Cell viability regarding 5.5 (µg/ml) AgNPs solution in Figure 7 B.
it will be better if the authors can make it more concise.
Author Response
We thank the reviewers for their thoughtful and helpful comments. We respond to each point below; their suggestions have undoubtedly improved the paper, for which we are very grateful. All changes in the revised manuscript were in yellow highlighted.
- What are the components in the extraction of Jacobaea Maritima leaves, and their concentration?
Response: Thank you for your comment. Active compounds present in Jacobaea Maritima had already been identified by Voynikov et al (Voynikov, Y., et al., Chemophenetic Approach to Selected Senecioneae Species, Combining Morphometric and UHPLC-HRMS Analyses. Plants (Basel), 2023. 12), using Ultra-High-Performance Liquid Chromatography - High-Resolution Mass Spectrometry (UHPLC-HRMS), which are summarized in Table S1, in the Supplementary Materials Section. These compounds are classified as either flavonoids or hydroxybenzoic, hydroxycinnamic, and acylquinic acids along with their derivatives.
- Which component is involved in the reduction reaction with AgNo3, and what is the pathway of reaction?
Response: No exact mechanism is known for the biosynthesis of AgNPs using plants’ extracts. However, certain functional groups are identified for the reduction of Ag ions including hydroxyl groups attached to aromatic rings and carboxyl groups, which are present in the active compounds listed in Table S1. It is plausible that the Enol- keto tautomeric transformation of these compounds could release a reactive free electron that reduces silver ions to metallic silver (Page 4, Lines 135-145 in the revised manuscript).
- What are the concentrations of active extraction and silver nitrate in line 303-305?
Response: AgNO3 solution was prepared at a concentration of 1mM while the extract was prepared by mixing 25 gm of finely incised leaves with 300 mL of distilled water and the mixture was left to boil for 20 min, then the temperature was decreased to 50°C and kept for 25 min. Next, filtration was carried out to obtain the extract solution. Finally, a 1:1 ratio of AgNO3: extract solution was used (Page 11, Lines 348-353 in the revised manuscript).
- Why can the lower reaction temperature result in smaller AgNPs?
Response: It is already established that reaction temperature affects the formation and growth of AgNPs; the higher the temperature the larger the nanoparticle size, which agrees with the obtained results. One plausible explanation is that high temperatures accelerate the nucleation and growth of nanoparticles, which occurs via Ostwald ripening or coalescence to form larger particles that are more energetically stable (Page 4-5, Lines 153-165 in the revised manuscript).
- What component played as capping agents of resultant clusters, and why did the zeta potential turn from negative (-31.7) to positive (10.9)?
Response: It is advantageous to use plant extract to synthesize NPs as multiple capping agents can form a thin layer surrounding the NPs. Based on FTIR analysis, compounds containing carbonyl and phenolic groups along with aromatic rings can be found on the surface of NPs, which is evidence of the presence of flavonoids or acids extract along with their derivatives. Thereby, FTIR analysis revealed the efficient capping and stabilization of AgNPs (Page 7, Lines 218-224 in the revised manuscript).
We apologize for the typo mistake, the zeta potential value decreased from -31.7 to -10.9 mV. The smaller the nanoparticles the larger their surface area and thereby the density of capping agents increases on the surface. Owing to the enol- keto tautomeric transformation of compounds present in the plant extract, which releases a reactive free electron to reduce silver ions to metallic silver, most capping agents present in the solution will lose their charges and thereby reducing the zeta potential value of formed AgNPs. This is confirmed by FTIR analysis that shows the loss of OH stretching vibration at 3239 cm-1 for AgNPs while having stronger peaks at 1032 cm-1 and 1261 cm-1, which can be attributed to C-O stretching of carboxylic acid, esters, or ethers (Page 6, Lines 188-194 in the revised manuscript).
- The SEM image is not clear at all, using a TEM image is better to measure the particle size as in your previous paper.
Response: A higher resolution SEM image has been provided with a better visualization of the spherical shape of AgNPs (Figure 3, Page 6 in the revised manuscript), as unfortunately, we don’t have access to TEM imaging.
- I can’t find the column indicating Cell viability regarding 5.5 (µg/ml) AgNPs solution in Figure 7 B.
Response: The percentage of cell viability of 5.5 µg/mL concentration is 1.18% (very small column); therefore, the column does not appear clearly in Figure 7B. Moreover, it should be noted the error bars of all concentrations have been adjusted accordingly in both figures (7A & 7B).
Reviewer 3 Report
Comments and Suggestions for Authors
The authors have done a significant amount of work, and the most interesting part is the fact that this study targets two biomedical applications simultaneously through one material. There are some intricate details missing in this study and some of them are mentioned below. The authors need to address these in detail.
1. The authors need to discuss the progress of silver nanoparticles in antibacterial and tumor or cancerous studies in the introduction from both chemical and green synthesis as a support to this study because it is also a necessary part. The introduction must give 2-3 examples specifically to these areas of applications which is the objective of the study to answer why this study will be considered as progress. Some suggested references are:
a) https://doi.org/10.1080/21691401.2017.1337031
b) https://doi.org/10.3390/nano9091282
c) https://doi.org/10.1016/j.jrras.2015.01.007
d) https://doi.org/10.36468/pharmaceutical-sciences.646
2. The authors need to show statistical significance for data provided in this study specifically for the results related to the antibacterial and anticancer part. Statistical significance will give the effectivity of the work and will grab the reader’s attention towards a particular experimental setup and its reproducibility/validation.
3. The authors need to provide a higher-clarity SEM image for the silver nanoparticles. Authors might consider gold-palladium sputter coating for increasing conductivity and better contrast images.
4. In section 2.3, how will the author explain the change in particle size and shift of zeta potential to that extent when the Ag particles were synthesized at 4ËšC? Reference 34 does not discuss the shift in zeta potential and is not the related explanation here based on their observed study. Salayova et al. did not mention any relation between particle size and zeta potential in their article.
Comments on the Quality of English LanguageSome of the complex sentences can be made shorter for better clarity.
Author Response
We thank the reviewers for their thoughtful and helpful comments. We respond to each point below; their suggestions have undoubtedly improved the paper, for which we are very grateful. All changes in the revised manuscript were in yellow highlighted.
- The authors need to discuss the progress of silver nanoparticles in antibacterial and tumor or cancerous studies in the introduction from both chemical and green synthesis as a support to this study because it is also a necessary part. The introduction must give 2-3 examples specifically to these areas of applications which is the objective of the study to answer why this study will be considered as progress. Some suggested references are:
- a) https://doi.org/10.1080/21691401.2017.1337031
- b) https://doi.org/10.3390/nano9091282
- c) https://doi.org/10.1016/j.jrras.2015.01.007
- d) https://doi.org/10.36468/pharmaceutical-sciences.646
Response: Thank you for your suggestion. The following paragraph discusses the progress of AgNPs in antibacterial and cancerous studies and has been added to the introduction (Page 2, Lines 72-84 in the revised manuscript).
(Recently, AgNPs have gained significant interest in the biomedical field due to their outstanding characteristics as a potential therapeutic approach in the treatment of different types of diseases such as bacterial and fungal infections, inflammation and cancers. The green synthesis of AgNPs from natural sources has been demonstrated to involve significantly in reducing the progression of human hepatic cancer cells. Applying the extract of Punica granatum leaf, an ancient fruit belonging to the Punicaceae family, is reported to have anti-oxidant activity and free radical-scavenging potency suggesting promising and useful applications in the field of biomedical research. Moreover, the green synthesis of AgNPs from banana peel extract (BPE) which is agricultural waste material and used as a reducing and capping agent revealed a cost-effective, non-toxic, and eco-friendly approach for the synthesis of AgNPs. The study showed a synergistic effect on the antimicrobial activity of the standard antibiotic levofloxacin against Gram-positive and Gram-negative bacteria).
- The authors need to show statistical significance for data provided in this study specifically for the results related to the antibacterial and anticancer part. Statistical significance will give the effectivity of the work and will grab the reader’s attention towards a particular experimental setup and its reproducibility/validation.
Response:
Thank you for your comment. Statistical differences are very crucial for studies which compare 2 or more groups together. For the antibacterial test, the sudden drop in the OD indicates the inhibition of bacterial growth. There is no need to compare the groups together because the presence of bacteria means there will be represented by tall columns and the inhibition of growth will be represented by the sort columns, so it is not applicable here. For the cytotoxicity study, here we just want to demonstrate that the leaf extract is not toxic to the cancer cells even in high concentrations, and the AgNPs possess the anticancer effect. The important finding here is the estimation of the IC50 against both cancer cell lines, which can be easily estimated now, so again it is not applicable for this study as well.
- The authors need to provide a higher-clarity SEM image for the silver nanoparticles. Authors might consider gold-palladium sputter coating for increasing conductivity and better contrast images.
Response: A higher resolution SEM image is provided with a better visualization of the spherical shape of AgNPs (Figure 3, Page 6 in the revised manuscript). Your kind suggestion will be considered in our future work.
- In section 2.3, how will the author explain the change in particle size and shift of zeta potential to that extent when the Ag particles were synthesized at 4ËšC? Reference 34 does not discuss the shift in zeta potential and is not the related explanation here based on their observed study. Salayova et al. did not mention any relation between particle size and zeta potential in their article.
Response: It is already established that reaction temperature affects the formation and growth of AgNPs; the higher the temperature the larger the nanoparticle size, which agrees with the obtained results. One plausible explanation is that high temperatures accelerate the nucleation and growth of nanoparticles, which occurs via Ostwald ripening or coalescence to form larger particles that are more energetically stable. The smaller the nanoparticles the larger their surface area and thereby the density of capping agents increases on the surface. Owing to the enol- keto tautomeric transformation of compounds present in the plant extract, which releases a reactive free electron to reduce silver ions to metallic silver, most capping agents present in the solution will lose their charges and thereby reducing the zeta potential value of formed AgNPs. This is confirmed by FTIR analysis that shows the loss of OH stretching vibration at 3239 cm-1 for AgNPs while having stronger peaks at 1032 cm-1 and 1261 cm-1, which can be attributed to C-O stretching of carboxylic acid, esters, or ethers (Page 5-6, Lines 187-194 in the revised manuscript).
We have referenced Salayova et al (reference 39 in the revised manuscript), due to their similar AgNP size and zeta potential results. I highlighted my explanation of the shift in zeta potential with previous literature with reference under Dynamic Light Scattering results and discussion section).
Reviewer 4 Report
Comments and Suggestions for Authors
The research work presented by Tawfik et al. is well designed but there are major issues that should be addressed before full acceptance of the manuscript:
1) The authors synthesized three types of silver nanoparticles using Jacobaea maritima leaf extracts through different experimental conditions (100 ºC, 22ºC and 4ºC). However, only one of them was fully characterized. Why?
2) I assume that the study was performed using the AgNPs obtained at 4 ºC because they were the ones mentioned in the Abstract and Conclusion sections, but it is not clearly mentioned in the main text. A discussion stating the choice of this AgNPs should be done.
3) The IR spectrum of AgNPs is very poor and thus, conclusions about the bands at 1653 and 2079 cm-1 made by the authors are doubtful.
4) Phares in lines 226-230 are very confusing. It is stated that the results illustrate that the sensitivity of the bacteria stains are mainly due to the AgNP’s and not because silver ions, but the latter (AgNO3) are more active. The authors also claim that AgNPs with small sizes have higher surface area, hence release more silver ions than those of the low surface area resulting in antibacterial toxicity. As so, one can conclude that the activity is always is due to the Ag+ ions present in solution. How can the authors correlate the size of the AgNPs with antibacterial activity as only one type of AgNPs was tested? Please clarify these issues.
5) I consider the paragraph in lines 258-273 too much speculative because no mechanistic studies were performed in this study. I suggest removing it.
6) The antimicrobial results need to be carefully revised. The negative control corresponds to the bacteria growing in the chosen medium (no inhibition, i. e. OD = 1) and the positive control to a compound with known activity (i. e. AgNO3 in this study). It is senseless to present the media only results and consider them the negative because they will always present an OD = 0, which stands for the positive control. Figures presented in SI must be modified accordingly.
Minor issues:
Title: Replace “…its…” by “…their…”.
Title: Change “…Maritima…” by “…maritima…”.
Line 22/23: Replace “…was through…” by “…was performed by…”.
Lines 121-130: Merged this part of the text with section 2.1.
Line 157: Remove “absorbed”.
Line 159: Remove “formation”.
Line 188: Replace “…even at low concentrations, i.e., ≤ 1.4 µg/mL. However, more...” by “…at a dose ≥ 1.4 µg/mL. More…”.
Lines 189-191: Remove “….when…control….”
Line 200: Replace “…had inhibited…” by “…generally inhibits…”.
Lines 218-219: Merge the first phrase with the Introduction section.
Line 224: Remove “MIC”.
Line 225: Replace “…had an MIC of…” by “…displayed a MIC value of…“.
Table 2: Remove the first column.
Line 237 and 240: Replace “…one-fold less…” by “…half…”.
Line 239: Replace “…two-fold lower…” by “…a quarter lower…”.
Line 239: Remove “However”.
In all document carefully check the systematic name of plant species that should be in italic style and bear in mind that the IR spectra reveals absorption bands rather than peaks.
Comments on the Quality of English LanguageMinor editing of English language required.
Author Response
We thank the reviewers for their thoughtful and helpful comments. We respond to each point below; their suggestions have undoubtedly improved the paper, for which we are very grateful. All changes in the revised manuscript were in yellow highlighted.
- The authors synthesized three types of silver nanoparticles using Jacobaea maritime leaf extracts through different experimental conditions (100 ºC, 22ºC and 4ºC). However, only one of them was fully characterized. Why?
Response: Thank you for your comment. Based on previous anti-microbial results of biosynthesized AgNPs using plants, a smaller size around 50 nm is much preferable for the antibacterial activity. Therefore, we tried to optimize different factors (reaction temperature, reaction time, precursor concentration, and extract concentration), in the synthesis process to obtain a similar approximate size. Hence, DLS analysis was mainly performed for the characterization of the developed NPs as size optimization was our first target. Once the optimal size was reached, full characterization was performed using SEM, XRD, and FTIR analyses.
- I assume that the study was performed using the AgNPs obtained at 4 ºC because they were the ones mentioned in the Abstract and Conclusion sections, but it is not clearly mentioned in the main text. A discussion stating the choice of this AgNPs should be done.
Response: The AgNPs obtained in temperature decreased gradually from 22°C to 4°C, over 72 hour period, not at 4°C (Page 5, Lines 179-182 in the revised manuscript).
- The IR spectrum of AgNPs is very poor and thus, conclusions about the bands at 1653 and 2079 cm-1made by the authors are doubtful.
Response: Thank you for your comment. We tried to improve the resolution of the figure in the revised manuscript. We appreciate that the resolution is not the best, as it is common for biosynthetic materials and natural compounds; hence, it will be difficult to have no noise similar to the chemically synthesized and pure forms of drugs and other compounds.
- Phares in lines 226-230 are very confusing. It is stated that the results illustrate that the sensitivity of the bacteria stains are mainly due to the AgNP’s and not because silver ions, but the latter (AgNO3) are more active. The authors also claim that AgNPs with small sizes have higher surface area, hence release more silver ions than those of the low surface area resulting in antibacterial toxicity. As so, one can conclude that the activity is always is due to the Ag+ions present in solution. How can the authors correlate the size of the AgNPs with antibacterial activity as only one type of AgNPs was tested? Please clarify these issues.
Response: The section has been removed and this statement was moved to below the Table (Page 10, Lines 300-302 in the revised manuscript) ‘Owing to the smaller size of AgNPs (i.e., 37 ± 10 nm), higher surface area and more silver ions will be released, resulting in antibacterial activity’
- I consider the paragraph in lines 258-273 too much speculative because no mechanistic studies were performed in this study. I suggest removing it.
Response: This paragraph has been removed.
- The antimicrobial results need to be carefully revised. The negative control corresponds to the bacteria growing in the chosen medium (no inhibition, i.e. OD = 1) and the positive control to a compound with known activity (i.e. AgNO3in this study). It is senseless to present the media only results and consider them the negative because they will always present an OD = 0, which stands for the positive control. Figures presented in SI must be modified accordingly.
Response:
In our study, mainly we followed the guidelines of the CLSI for microdilution method (CLSI-M07, 2015) that we use regularly in our lab and previously published papers; further a reference article that conducted a similar study of using AgNPs as an antibacterial agent (more details in https://doi.org/10.3389/fmicb.2022.820048). It is apparent in the guideline: the positive control (containing only bacteria) to ensure optimal bacterial growth and to compare it with the inhibition of our tested agent in this case AgNPs, where the growth reduction is ≥ 80% considered as MIC. Also, we determine its inhibition by the naked eye. For the negative control (containing only media) ensure the 96-well plate is not contaminated. After that, we checked the bacterial inhibition using a UV-spectrophotometer (at 600nm) as demonstrated in the manuscript. After reviewing the biosynthesis of AgNPs, we have revised the manuscript and modified the data regarding AgNO3 which is not applicable in this study as the last step of the protocol for the biosynthesis of AgNPs (More details on https://doi.org/10.3390/nano11020487), the remaining solution of AgNO3 is washed several times by deionized water. In all, we don’t have to include AgNO3 part of the MIC assay since the residual organic compounds of AgNO3 are removed during the biosynthesis and no residual of AgNO3 that could interfere with the bacterial inhibition of newly biosynthesized AgNPs. However, we decided to include its results in the Supplementary Materials Section Figures S1-S8 to show its effectiveness against different bacterial strains to be further investigated in future studies.
Minor issues:
Title: Replace “…its…” by “…their…”.
Response: The title has been changed accordingly.
Title: Change “…Maritima…” by “…maritima…”.
Response: The title has been changed accordingly.
Line 22/23: Replace “…was through…” by “…was performed by…”.
Response: This word has been replaced.
Lines 121-130: Merged this part of the text with section 2.1.
Response: This part has been merged.
Line 157: Remove “absorbed”.
Response: This word has been removed.
Line 159: Remove “formation”.
Response: This word has been removed.
Line 188: Replace “…even at low concentrations, i.e., ≤ 1.4 µg/mL. However, more...” by “…at a
dose ≥ 1.4 µg/mL. More…”.
Response: This has been replaced
Lines 189-191: Remove “….when…control….”
Response: This word has been removed.
Line 200: Replace “…had inhibited…” by “…generally inhibits…”.
Response: This has been replaced.
Lines 218-219: Merge the first phrase with the Introduction section.
Response: Thank you for the suggestion. We preferred to leave the statement as an introductory statement for this paragraph.
Line 224: Remove “MIC”.
Response: This word has been removed.
Line 225: Replace “…had an MIC of…” by “…displayed a MIC value of…“.
Response: This has been replaced.
Table 2: Remove the first column.
Response: The column has been removed.
Line 237 and 240: Replace “…one-fold less…” by “…half…”.
Response: This has been replaced.
Line 239: Replace “…two-fold lower…” by “…a quarter lower…”.
Response: This has been replaced.
Line 239: Remove “However”.
Response: This word has been removed.
In all document carefully check the systematic name of plant species that should be in italic style and bear in mind that the IR spectra reveals absorption bands rather than peaks
Response: All plants' name has been checked and adjusted accordingly. Thank you!
Round 2
Reviewer 2 Report
Comments and Suggestions for Authors
no more comments
Reviewer 4 Report
Comments and Suggestions for Authors
The authors have adressed all questions raised. There are still some phrases that give repited ideas along the manuscript that would be removed for clarity but it does not harm it acceptance.
Comments on the Quality of English LanguageMinor editing of English language required.